# Role of Vitrectomy in Nontractional Refractory Diabetic Macular Edema

**DOI:** 10.3390/jcm12062297

**Published:** 2023-03-15

**Authors:** Stefano Ranno, Stela Vujosevic, Manuela Mambretti, Cristian Metrangolo, Micol Alkabes, Giovanni Rabbiolo, Andrea Govetto, Elisa Carini, Paolo Nucci, Paolo Radice

**Affiliations:** 1Ophthalmology Department, Ospedale di Circolo e Fondazione Macchi, ASST Sette Laghi, 21100 Varese, Italy; 2Department of Biomedical, Surgical and Dental Sciences, University of Milan, 20122 Milan, Italy; 3Eye Clinic, IRCCS MultiMedica, 20138 Milan, Italy; 4Eye Clinic, University Hospital Maggiore della Carità, 28100 Novara, Italy; 5Ophthalmology Department, Ospedale A. Manzoni, 23900 Lecco, Italy

**Keywords:** diabetic macular edema, pars plana vitrectomy, ILM peeling

## Abstract

Background: Currently, the gold standard of diabetic macular edema (DME) treatment is anti-vascular endothelial growth factor (VEGF) injections, although a percentage of patients do not respond optimally. Vitrectomy with or without internal limiting membrane (ILM) peeling is a well-established treatment for DME cases with a tractional component while its role for nontractional cases is unclear. The aim of this study is to evaluate the role of vitrectomy with or without ILM peeling in nontractional refractory DME. Methods: We performed a retrospective review of twenty-eight eyes with nontractional refractory DME treated with vitrectomy at San Giuseppe Hospital, Milan, between 2016 and 2018. All surgeries were performed by a single experienced vitreoretinal surgeon. In 43.4% of cases, the ILM was peeled. Best corrected visual acuity and optical coherence tomography (OCT) scans were assessed preoperatively and at 6, 12, and 24 months post-vitrectomy. Results: The mean central macular thickness improved from 413.1 ± 84.4 to 291.3 ± 57.6 μm at two years (*p* < 0.0001). The mean logarithm of the minimum angle of resolution logMAR best-corrected visual acuity (BCVA) improved after two years, from 0.6 ± 0.2 to 0.2 ± 0.1 (*p* < 0.0001). We found no difference between ILM peeling vs. no ILM peeling group in terms of anatomical (*p* = 0.8) and visual outcome (*p* = 0.3). Eyes with DME and subfoveal serous retinal detachment (SRD) at baseline had better visual outcomes at the final visit (*p* = 0.001). Conclusions: We demonstrated anatomical and visual improvement of patients who underwent vitrectomy for nontractional refractory DME with and without ILM peeling. Improvement was greater in patients presenting subretinal fluid preoperatively.

## 1. Introduction

Diabetic retinopathy is the main cause of visual impairment in working-age populations of industrialized countries. Vision loss may result from several mechanisms, but the most common cause is diabetic macular edema (DME) [1].

Treatment of DME has evolved in the last ten years [2]. While focal or grid macular argon laser photocoagulation used to be the gold standard treatment in the past, [3] nowadays the indication for laser therapy is limited specifically to non-center-involved DME [4]. Currently, the most commonly used therapeutic options include intravitreal administration of anti-vascular endothelial growth factor (anti-VEGF) agents and corticosteroids [5,6]. Although anti-VEGF treatment has been shown to significantly improve visual function outcomes in patients with DME, there is still a significant proportion of patients who are non-responders (23%) [7].

In some patients, a mechanical component, due to vitreomacular traction or tractional epiretinal membrane (ERM), may contribute to the pathogenesis of DME. In such cases, vitrectomy and surgical relaxation of the traction have been suggested as effective treatment methods for DME [8]. However, the use of this procedure for the treatment of nontractional DME remains controversial. The rationale behind utilizing pars plana vitrectomy (PPV) for treating nontractional DME can be supported by the improvement of vitreous oxygenation and by the reduction of vitreous VEGF and cytokine levels [9]. The aim of this study was to report on the long-term (two years) morphologic and visual outcomes of patients with nontractional refractory DME who underwent PPV.

## 2. Materials and Methods

We performed a retrospective chart review of the records of all patients with nontractional refractory DME who were surgically treated at San Giuseppe Hospital, Milan, between January 2016 and December 2018, and had follow-up data that covered a period of two years. Refractory DME was defined as visual acuity (VA) improvement of five letters or less (including any vision loss), or a reduction in central macular thickness (CMT) inferior to 20%, as measured with spectral domain optical coherence tomography (SD-OCT), after at least six intravitreal anti-VEGF injections of aflibercept (Eylea; Bayer, Leverkusen, Germany) over 8 months, allowing for a loading dose of 5 monthly injections and a 6th injection after 2 months [10].

The main exclusion criteria were presence of any other retinal vascular disease, presence of ERM, vitreomacular traction syndrome, age-related macular degeneration, and history of previous vitrectomy or retinal detachment. We also excluded patients with proliferative diabetic retinopathy (PDR).

The best-corrected visual acuity (BCVA) values and the SD-OCT (Cirrus HD-OCT, Carl Zeiss Meditec, Inc., Dublin, CA, USA) results were assessed preoperatively and at 6, 12, and 24 months post-vitrectomy. BCVA was measured using the Early Treatment Diabetic Retinopathy Study (ETDRS) charts, and the letter score was converted to logarithm of minimum angle of resolution (logMAR) units for statistical purposes.

The CMT values were obtained from the central subfield of the macular thickness map. The following patterns of DME were evaluated on SD-OCT B-scan images: cystoid DME, diffuse DME, and presence of subfovealneuroretinal detachment [11].

### 2.1. Surgical Procedure 

All surgeries were performed by a single surgeon (S.R.). Regional anesthesia was achieved in all patients by means of a peribulbar block. For each patient, a core vitrectomy was performed using the 25G Stellaris posterior and cataract PC instrument (Bausch + Lomb; Vimodrone, Milan, Italy). The posterior vitreous was separated from the retina by active aspiration using a vitrectomy probe, and any visible vitreous strands which adhered to the retina were removed. Intravitreal triamcinolone (40 mg/mL, Intracinol; Farmigea, Pisa, Italy) was used in all cases to facilitate visualization and removal of the adherent posterior cortical vitreous. Internal limiting membrane (ILM) peeling was not systematically performed. When performed, trypan blue 0.15% and brilliant blue G were used (Dual-Blu; Dorc, Zuidland, The Netherlands) to stain and then remove the ILM in a circular area of two to three optical disc diameters around the fovea. The standard surgical procedure used in 2016 did not involve ILM peeling, whereas ILM peeling was included in surgeries performed from 2017 onwards. All patients were pseudophakic at the time of vitreoretinal surgery. No tamponade was used except for 3 patients in which iatrogenic breaks required air tamponade.

### 2.2. Statistical Analysis 

The unpaired Student’s t-test was used to compare continuous data. The paired Student’s *t*-test was used to statistically evaluate comparisons between preoperative and postoperative outcomes when appropriate. A *p*-value less than 0.05 was considered to be statistically significant. Normality of the data was confirmed by a Shapiro–Wilktest.

For binary outcomes, Fisher’s exact test was used for intergroup comparisons of proportions when appropriate. Univariate comparisons by linear regression were performed to identify associations between final best-corrected visual acuity BCVA variations and other variables under study. All statistical analyses were performed using SPSS for Windows version 20.0 (IBM Corp., Armonk, NY, USA).

## 3. Results

Twenty-eight eyes of twenty-eight patients with nontractional refractory DME, who were treated with vitrectomy and were followed up for two years, were analyzed. No patient was lost during the follow-up. The subjects included 16 men and 12 women, with a mean age of 73.6 + 9.4 years. The baseline characteristics of the patients are shown in Table 1. The mean number of intravitreal treatments administered before surgery was 8 ± 1 (range 6 to 10).

Mean preoperative CMT was 413.1 ± 84.4 μm, and Shapiro–Wilk tests did not show a significant departure from the normality (*p* = 0.17).

The mean CMT improved significantly from 413.1 ± 84.4 μm at baseline to 291.3 ± 57.6 μm at two years (*p <* 0.0001), with a significant reduction recorded even at six months (*p <* 0.0001) and 12 months (*p <* 0.0001). The mean logMAR BCVA significantly improved after two years, from 0.6 ± 0.2 to 0.2 ± 0.1 (*p <* 0.0001), with a significant improvement recorded at six months (*p <* 0.0001) and 12 months (*p <* 0.0001) (Table 2).

The BCVA at two years was influenced by the duration of diabetes (R^2^ = 0.61) and DME (R^2^ = 0.71) (Figure 1 and Figure 2).

The ILM was peeled in 13 eyes (43.4%) but was not peeled in 15 eyes (53.6%). There was no statistically significant difference between the two groups in terms of reduction in macular thickness (*p* = 0.8) and final visual outcome (*p* = 0.3).At the two-year follow-up, an epiretinal membrane (ERM) was observed in seven of the eyes which did not undergo ILM peeling (46.7%), whereas none of the eyes which underwent ILM peeling developed an ERM (*p* = 0.005).

By analyzing OCT patterns, 12 out of 28 patients (42.85%) had DME with subfoveal serous retinal detachment (SRD) at baseline. This subset of patients had better visual results at two years compared to patients who did not have subretinal fluid(SRD) (0.15 ± 0.1 and 0.33 ± 0.1, respectively) (*p* = 0.001) (Figure 3).

No eye received any adjuvant treatment during the first year after surgery; however, seven eyes (25%) received intravitreal corticosteroid injections in the second year for the treatment of worsening DME, of which four once only and three had two injections.

None of the eyes that received extra treatment belonged to the subgroup of patients with SRD.

Of the 28 patients treated, there were 2 cases of retinal detachment in which 1 patient is among those who received 2 injections of corticosteroids and 1 case of vitreous hemorrhage during the follow-up period that required a second intervention. Among intraoperative complications we had three cases of iatrogenic breaks that were treated with intraoperative laser and air tamponade. Of the 28 patients therefore 16 did not have any intraoperative or postoperative complications and did not need additional treatment during the follow-up.

## 4. Discussion

The aim of this study was to report on the long-term morphologic and visual outcomes of patients with nontractional refractory DME who underwent PPV. The role of vitrectomy in cases of nontractional DME is a controversial topic. For patients with vitreomacular traction or a tractional ERM, vitrectomy is recommended because anteroposterior and tangential vitreomacular traction plays a key role in the genesis of macular edema [12]. Nevertheless, different authors have also described vitrectomy to be beneficial even when macular traction is absent [13,14,15,16].In the present study, our results showed a reduction in macular thickness two years after PPV in patients with refractory nontractional DME, indicating a significant anatomical improvement. The anatomic outcomes recorded in the present study are in line with those of almost all previous studies on this topic [13,14,15,16]. The exact mechanism by which vitrectomy may improve DME has yet to be understood, however, there are different mechanisms that may justify these results. It is well known that vitrectomy increases the levels of oxygen delivered to the inner retina [17] and improves perifoveal capillary blood flow [18]. Oxygen has been shown to suppress VEGF, which in turn alters capillary permeability leading to DME [19].Therefore, an increase in oxygen concentration would act as a potent anti-VEGF agent. Furthermore, it has been proven that the removal of the vitreous reduces levels of histamine, VEGF, and free radicals in the preretinal space [20].

In the present study, our results showed improved visual acuity in addition to anatomical improvement. Some studies reported greater visual improvement, but the results on this topic are very heterogeneous. BCVA gains greater or equal to two ETDRS lines have been reported in around 50% of cases(range, 20–90%) in the available literature [11,21,22,23,24,25,26]. On the other hand, several studies did not report any visual improvement despite results that indicated anatomical improvement [9,27,28,29]. There may be various reasons for the differences in these results, such as the heterogeneity of the baseline characteristics of the patients who underwent surgery and the heterogeneity of the procedures performed. In a few studies, the patients were pseudophakic before they underwent PPV; in others, cataract surgery was performed at the same time or during a second procedure. Since cataract influences visual acuity, it would be difficult to quantify the visual gain achieved due to PPV. We bypassed the effect of this confounding factor by including only patients who had previously undergone cataract surgery.

Furthermore, a factor that hinders functional improvement is retinal damage, especially in refractory and long-standing DME. In such cases, despite good anatomical results, the visual gain is often limited because of pre-existing damage to the outer retina layers and the external limiting membrane. Therefore, an extremely relevant factor for functional recovery could be the timing of the surgery. To the best of our knowledge, there are few studies in the available literature in which PPV was performed early in treatment-naive patients with positive outcomes [30,31]. In these studies, time was considered a predictor for good functional treatment response. According to Iglicki et al., [30] for every day that surgery is postponed, the chances of gaining more than five letters decrease by 1.8%. In the present study, all the patients had refractory, long standing DME, and this may have affected the extent of their visual gain.

After performing a subgroup analysis based on the OCT classification of DME, we found a greater visual improvement in patients who had SRD at baseline. Our results are in line with those of recent studies which suggest that the effectiveness and results of DME therapy can be determined from preoperative OCT patterns [30,32].Ichiyama et al. [32] also noticed a higher improvement in the vision of DME patients who had SRD. Visual outcomes are worse in patients with a mixed form of DME and are very poor in patients with cystoid macular edema or in patients with sponge-like diffuse retinal thickening.

The relationship between the effectiveness and results of DME therapy and OCT patterns could be explained by subtle pathogenetic differences. The origin of SRD in DME is unclear, although impairment of the retinal pigment epithelium (RPE has been suggested as the cause for SRD. Studies have demonstrated the decreased ability of the RPE to pump fluid in hypoxic conditions [33]. PPV may improve the oxygenation of RPE, which in turn increases the ability of the RPE to pump fluid. This may explain the better results observed in DME patients who had SRD. On the contrary, persistent sub-retinal fluid may damage photoreceptor cells. Consequently, if SRD can be considered an anatomical predictor of good visual outcomes in patients treated with PPV it would stand to reason that vitrectomy should be considered early on in refractory DME.

However, this idea is debatable as other studies have also shown positive outcomes when using intravitreal dexamethasone implants compared to anti-VEGF to treat DME patients with SRD. As SRD is considered an inflammatory biomarker, the results have been linked to a more efficient anti-inflammatory effect of corticosteroids. [34] Vitrectomy may have a similar effect, by removing a large reservoir of pro-inflammatory cytokines. [35] Further studies are needed to investigate and clarify this idea.

Another controversial notion is the advantage of ILM peeling in treating patients with nontractional DME. The available literature regarding this topic is contradictory. A recent meta-analysis showed no advantage in treating nontractionaldiabetic macular edema with ILM peeling compared with treatment without ILM peeling, [36] whereas some authors suggest better anatomical and functional results with ILM peeling [21,37]. In the present study, ILM peeling was performed on 46% of patients in the PPV group during the PPV procedure, whereas ILM peeling was not performed on 54% of the patients. When we analyzed these two groups separately, we found no differences in their visual acuity outcomes and anatomical improvement. As expected, the only significant difference was a lower rate of occurrence of ERM in the ILM peeling group after two years. ILM peeling has been shown to accelerate the resolution of macular edema, [21] possibly because it eliminates tangential tractional forces completely. The composition of ILM is altered in diabetes, due to an overexpression of collagen, fibronectin, and laminin, resulting in a thicker ILM compared to that of non-diabetic eyes [38]. All of these factors have been hypothesized to alter the fluid dynamics between the vitreous and the retina, justifying ILM peeling in diabetic eyes. Nowadays, however, the role of ILM peeling in the treatment of nontractional DME is controversial, and there is no substantial available data to clarify the efficacy of this procedure.

The present study has some limitations. The number of patients included is small, and no control group was included in the study. Additionally, we did not include patients that had been treated with intravitreal corticosteroids before vitrectomy. Moreover, we did not take into account the effect of peripheral argon laser treatment in the worsening of macular edema and epiretinal membrane formation. The retrospective design of the study is another notable limitation. Although the role of vitrectomy in DME has been evaluated in other studies, we have focused on patients who do not respond to anti-VEGF and demonstrated its usefulness in some DME patterns. To clarify the role of PPV in nontractional DME, large randomized controlled clinical trials that involve the comparison of the surgical approach to intravitreal treatment options are needed.

## Figures and Tables

**Figure 1 jcm-12-02297-f001:**
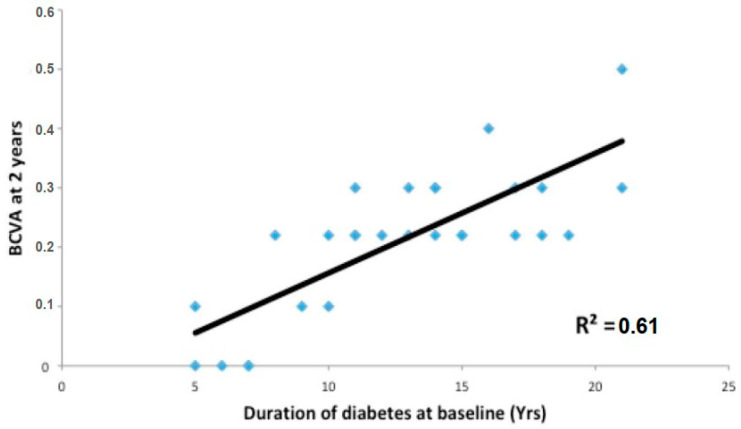
Pearson’s linear correlation between diabetes duration and final vision. BCVA. Best corrected visual acuity; Yrs, years.

**Figure 2 jcm-12-02297-f002:**
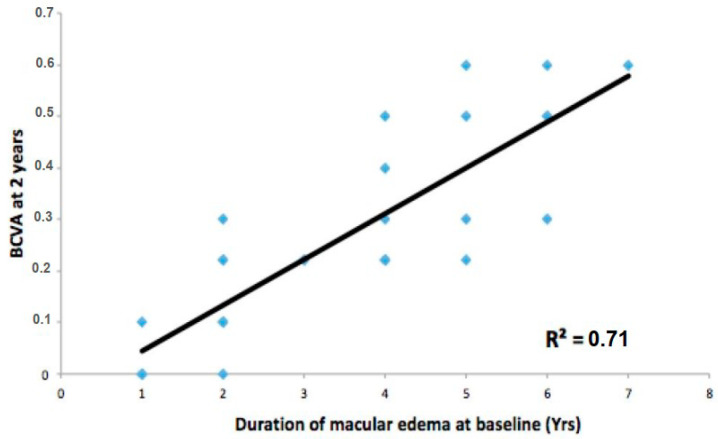
Pearson’s linear correlation between diabetes macular edema duration and final vision.

**Figure 3 jcm-12-02297-f003:**
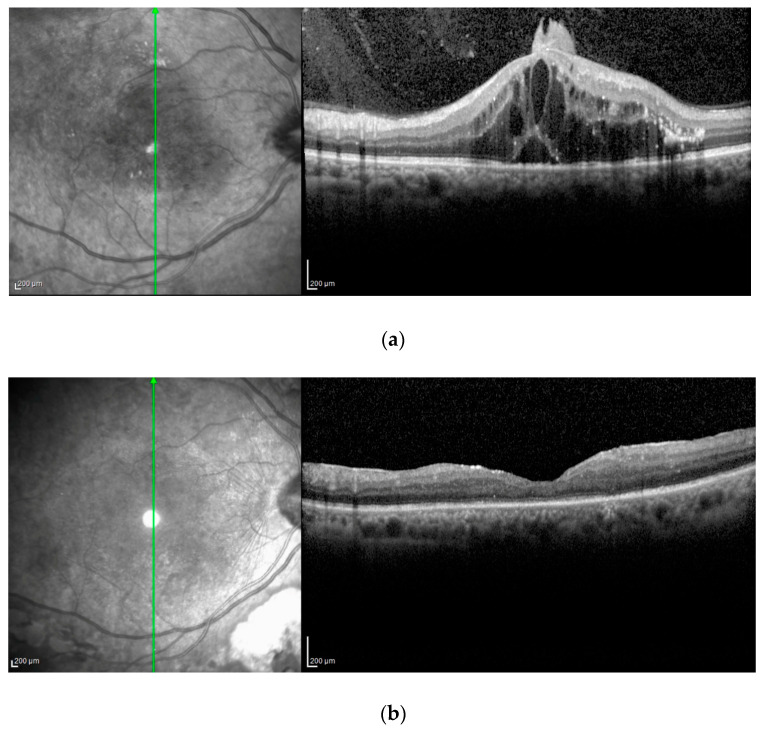
(**a**): Severe diabetic macular edema (DME) with subretinal fluid (SRF) and nodular epiretinal gliosis; (**b**): the same case two years after vitrectomy with complete resolution of DME.

**Table 1 jcm-12-02297-t001:** Baseline characteristics of the study groups.

	Patients Undergoing PPV (*n* 28)
Age (years)	73.6 ± 9.4
Sex (M/F)	16/12
HbA1c, %	6.7 ± 1.9
Diabetes duration (years)	12.7 ± 4.6
Macular Edema duration (years)	3.5 ± 1.8
Visual acuity at baseline (logMAR)	0.6 ± 0.2
Central macular thickness at baseline (microns)	413.1 ± 84.4

Characteristics are expressed by mean ± SD. PPV, pars plana vitrectomy; *n*, number

**Table 2 jcm-12-02297-t002:** BCVA and CMT at baseline and 2 years follow-up.

	Baseline	6 Months	*p*	12 Months	*p*	2 Years	*p*
BCVA	0.6 ± 0.2	0.3 ± 0.1	<0.0001	0.2 ± 0.1	<0.0001	0.2 ± 0.1	<0.0001
CMT	413.1 ± 84.4	308.4 ± 64.3	<0.0001	297.3 ± 48.1	<0.0001	291.3 ± 57.6	<0.0001

Data are expressed by mean ± SD.

## Data Availability

The data presented in this study are available on request from the corresponding author. The data are not publicly available due to privacy.

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
