# Peer review of "Role of Vitrectomy in Nontractional Refractory Diabetic Macular Edema"

_jcm, 2023, doi:10.3390/jcm12062297_

Round 1

Reviewer 1 Report

Ranno et al present a small study on 28 eyes that underwent vitrectomy for refractory DME without tractional components. The authors found improved BCVA and OCT CMT at 2 years. The strength of the study includes its 2-year follow up and good patient outcomes. The major limitations of this study is that there is a wealth of prior papers on this topic with much larger sample sizes and similar results, the early PPV intervention is not standard of care before trying other approved agents/lasers, and lack of information regarding complications and loss to follow up.

Detailed comments

1. The authors mention that their results are in line with references 12-15. For comparison, these references had sample sizes of 87 eyes, 73 eyes, 116 eyes, and 135 eyes. These studies are much larger than the 28 eyes that the authors present. What is truly new in this manuscript? Perhaps a paragraph in the discussion that highlights what this paper adds to the literature would be helpful.

2. The authors define Refractory DME: VA of improvement of 5 letters or less AND CMT reduction <20% after 6 injections. This is a debatable topic but seems like a reasonable definition, however, over what time period? Six injections over six months is much different than six injections over 2 years. Additionally, is this a single anti-VEGF injection type? Which anti-VEGF was used and was a second anti-VEGF attempted?

3. Refractory DME requiring surgical intervention is quite different than merely refractory DME. Standard of care (in the pre-faricimab era) among most practitioners would include 3 anti-VEGF injections, followed by 3 ant-VEGF injections of a different type before concluding that a patient is non-responsive to anti-VEGF. From this point, most practitioners would attempt either corticosteroids and/or laser before considering PPV for non-tractional DME. Therefore, this manuscript reports an artificial situation that is unlikely to be replicated by the majority of retina providers.

4. The authors make no mention of patients who are lost to follow up. Did all 28 patients truly complete 2 years? If patients did not complete 2 years, were they not included? If so, this would create quite a significant source of bias in the results. If patients who had poor outcomes did not make return visits, then they would not be included in the study, biasing it toward better outcomes.

5. The authors make no note of surgical complications. Were there any? PPV in diabetes with the hyaloid attached can be a very tricky procedure and retinal breaks are a serious risk. How many iatrogenic breaks were made?

6. How many post-operative anti-VEGF injections, steroid injections, or lasers were given? It states that 7 eyes received corticosteroids, just once? How many total injections?

Author Response

  1. The authors mention that their results are in line with references 12-15. For comparison, these references had sample sizes of 87 eyes, 73 eyes, 116 eyes, and 135 eyes. These studies are much larger than the 28 eyes that the authors present. What is truly new in this manuscript? Perhaps a paragraph in the discussion that highlights what this paper adds to the literature would be helpful.

RESPONSE: Reference 12-15 evaluated the effectiveness of PPV in diabetic macular edema. In our study we only analyzed the effectiveness of PPV in the subgroup of patients not responding to anti-vegf. Our aim was to assess whether in patients with edema refractory to medical treatment PPV could play a role. We added the clarification in the Discussion.

  1. The authors define Refractory DME: VA of improvement of 5 letters or less AND CMT reduction <20% after 6 injections. This is a debatable topic but seems like a reasonable definition, however, over what time period? Six injections over six months is much different than six injections over 2 years. Additionally, is this a single anti- VEGF injection type? Which anti-VEGF was used and was a second anti-VEGF attempted?

RESPONSE: We defined refractory DME as VA improvement of 5 letters or less and CMT reduction <20% after 6 injections over the course of 8 months maximum, allowing for a loading dose of 5 monthly injections and a 6th injection after 2 months.

  1. Refractory DME requiring surgical intervention is quite different than merely refractory DME. Standard of care (in the pre-faricimab era) among most practitioners would include 3 anti-VEGF injections, followed by 3 ant-VEGF injections of a different type before concluding that a patient is non-responsive to anti-VEGF. From this point, most practitioners would attempt either corticosteroids and/or laser before considering PPV for non- tractional DME. Therefore, this manuscript reports an artificial situation that is unlikely to be replicated by the majority of retina providers.

RESPONSE: As aflibercept was used we did not attempt a switch to a different anti-VEGF as, at the time, aflibercept had demonstrated similar to superior results in visual outcomes compared to ranibizumab and bevacizumab. The fact of having used only one type of anti-VEGF and not having used corticosteroids is a limitation of the study and is indicated in the discussion. However the study showed that in selected cases the PPV may be a treatment option.

  1. The authors make no mention of patients who are lost to follow up. Did all 28 patients truly complete 2 years? If patients did not complete 2 years, were they not included? If so, this would create quite a significant source of bias in the results. If patients who had poor outcomes did not make return visits, then they would not be included in the study, biasing it toward better outcomes.

RESPONSE: We didn’t exclude patients who got lost in the follow up because it would have been a bias, we didn’t have patients lost at the follow up. We added the data in the results

  1. The authors make no note of surgical complications. Were there any? PPV in diabetes with the hyaloid attached can be a very tricky procedure and retinal breaks are a serious risk. How many iatrogenic breaks were made?

RESPONSE: We had 2 retinal detachment and one vitreous hemorrhage during the follow-up period. Among intraoperative complications we had 3 cases of iatrogenic breaks. We added complication in the results.

  1. How many post-operative anti-VEGF injections, steroid injections, or lasers were given? It states that 7 eyes received corticosteroids, just once? How many total injections?

RESPONSE: 7 patients received injections of corticosteroids for a worsening of macular edema, of these 7, 4 once only and 3 two injections. We added it in the results

Reviewer 2 Report

Dear Editor,

Firstly, I want to thank you for giving me this opportunity. I have carefully read and reviewed this “Article” type of paper: “Role of vitrectomy in nontractional refractory diabetic macular edema”

The article is written about to evaluate the role of vitrectomy with or without ILM peeling in nontractional refractory DME.
The research content is in accordance with the aim and scope of the journal. Text language is clear. The subject and methods are interesting, although there are a few issues that the authors need to clarify and correct. If the authors consider the suggestions I have mentioned below, the manuscript will be significantly improved.

Good luck and success.

1)    First of all, “functional results” what did you do as a functional test? Microperimetry??? mERG?
If you haven't done an objective functional test, correct the entire manuscript as a “visual result/outcome” instead of a “functional result/outcome”.

2)    Wouldn't it be useful for objectivity to quantify whether the patients needed injections in this 2-year period? If yes, how many? Specify. If no patient needed even one injection during this 2-year period, clearly state this as well.

3)    "No tamponade was used." Was the end-of-case vitreous cavity left with air or BSS? Specify.

4)    Indicate whether there were any intra- or postoperative complications in the cases.

5)    "The mean CMT improved significantly from 413.1±84.4 μm at baseline to 291.3 ± 57.6 μm at two years (P< 0.0001), with a significant reduction recorded even at six months (P< 0.0001) and 12 months (P< 0.0001). The mean logMAR BCVA significantly improved after two years, from 0.6±0.2 to 0.2 ± 0.1 (P < 0.0001), with a significant improvement recorded 110 at six months (P< 0.0001) and 12 months (P< 0.0001) (Table 2) "
Remove the 6th month and 12th month if you are not going to include the 6th month and 12th month quantitative data in table 2 or in the manuscript.

6)    Can you name the hyperreflective material on the fovea in Figure 3a according to the nomenclature?

7)    What is meaning of VPP in Table 1?

8)    Last but not least, Hyperreflective spots such as SRD can also be seen as inflammatory markers.
I think the value of your research will increase if you can add hyperreflective spots to your work as you did for SRD.
You can tell if there is a relationship between the presence or amount of hyperreflective spots and PPV for these patient group.
I also recommend using these resources in the discussion:
1) https://doi.org/10.1097/iae.0000000000000912
2) https://doi.org/10.1177/1120672120962054

Kind Regards

Author Response

1) First of all, “functional results” what did you do as a functional test? Microperimetry??? mERG?
If you haven't done an objective functional test, correct the entire manuscript as a “visual result/outcome” instead of a “functional result/outcome”.
No microperimetry or mERG was performed. The manuscript has been amended accordingly.

RESPONSE: we made the suggested changes in the Manuscript

2) Wouldn't it be useful for objectivity to quantify whether the patients needed injections in this 2-year period? If yes, how many? Specify. If no patient needed even one injection during this 2-year period, clearly state this as well.

RESPONSE: we made 6 injections over the course of 8 months allowing for a loading dose of 5 monthly injections and a 6th injection after 2 months. During follow up 7 patients received injections of corticosteroids for a worsening of macular edema, of these 7, 4 once only and 3 two injections. We added it in the results

3)  "No tamponade was used." Was the end-of-case vitreous cavity left with air or BSS? Specify.

RESPONSE: we left the vitreous cavity in BSS in all patients except for 3 patients left in air because of iatrogenic breaks. We added in the results

4)  Indicate whether there were any intra- or postoperative complications in the cases.

RESPONSE: We had 2 retinal detachment and one vitreous hemorrhage during the follow-up period. Among intraoperative complications we had 3 cases of iatrogenic breaks. We added complications in the results.

5)  "The mean CMT improved significantly from 413.1±84.4 μm at baseline to 291.3 ± 57.6 μm at two years (P< 0.0001), with a significant reduction recorded even at six months (P< 0.0001) and 12 months (P< 0.0001). The mean logMAR BCVA significantly improved after two years, from 0.6±0.2 to 0.2 ± 0.1 (P < 0.0001), with a significant improvement recorded 110 at six months (P< 0.0001) and 12 months (P< 0.0001) (Table 2) "
Remove the 6th month and 12th month if you are not going to include the 6th month and 12th month quantitative data in table 2 or in the manuscript.

RESPONSE: We have all the follow up data that have been added in table 2.

6) Can you name the hyperreflective material on the fovea in Figure 3a according to the nomenclature?

RESPONSE: The figure description was updated with the correct terminology as described by Corvi et al. (doi:10.1016/j.oret. 2020.11.001)

7) What is meaning of VPP in Table 1?

This was corrected to PPV

8) Last but not least, Hyperreflective spots such as SRD can also be seen as inflammatory markers.
I think the value of your research will increase if you can add hyperreflective spots to your work as you did for SRD.
You can tell if there is a relationship between the presence or amount of hyperreflective spots and PPV for these patient group.

I also recommend using these resources in the discussion: 1) https://doi.org/10.1097/iae.0000000000000912
2) https://doi.org/10.1177/1120672120962054

RESPONSE: Thank you for your commentary. Unfortunately, hyperreflective spots analysis was not included in the study protocol at the time of ideation and it is not feasible at this time. We will surely keep it in mind for future studies.

Round 2

Reviewer 1 Report

Ranno et al present a small study on 28 eyes that underwent vitrectomy for refractory DME without tractional components. The authors found improved BCVA and OCT CMT at 2 years. The strength of the study includes its 2-year follow up and good patient outcomes. The authors answered many of concerns, however, their responses were not adequately transferred into the manuscript text. See details below.

Detailed comments

1a. My original comment: Refractory DME requiring surgical intervention is quite different than merely refractory DME. Standard of care (in the pre-faricimab era) among most practitioners would include 3 anti-VEGF injections, followed by 3 ant-VEGF injections of a different type before concluding that a patient is non-responsive to anti-VEGF. From this point, most practitioners would attempt either corticosteroids and/or laser before considering PPV for non-tractional DME. Therefore, this manuscript reports an artificial situation that is unlikely to be replicated by the majority of retina providers.

1b. RESPONSE: As aflibercept was used we did not attempt a switch to a different anti-VEGF as, at the time, aflibercept had demonstrated similar to superior results in visual outcomes compared to ranibizumab and bevacizumab. The fact of having used only one type of anti-VEGF and not having used corticosteroids is a limitation of the study and is indicated in the discussion. However the study showed that in selected cases the PPV may be a treatment option.

1c. My new comment: I do not see any mention regarding the use of a second anti-VEGF or laser therapy in the discussion limitations section.

2a. My original comment: The authors make no mention of patients who are lost to follow up. Did all 28 patients truly complete 2 years? If patients did not complete 2 years, were they not included?

2b. RESPONSE: We didn’t exclude patients who got lost in the follow up because it would have been a bias, we didn’t have patients lost at the follow up. We added the data in the results.

2c. My new comment: There is no mention of lost to follow up in the results as indicated. Please add this information.

3a. My original comment: How many post-operative anti-VEGF injections, steroid injections, or lasers were given? It states that 7 eyes received corticosteroids, just once? How many total injections?

3b. RESPONSE: 7 patients received injections of corticosteroids for a worsening of macular edema, of these 7, 4 once only and 3 two injections. We added it in the results

3c. My new comment: I see no mention of the number of injections per patient in the results, please add.

4. My new comment: was there any overlap between the 3 patients with post-operative complications, 3 iatrogenic breaks, and the 7 patients who required corticosteroid therapy after PPV? Of the 28 patients, how many patients had zero complications and needed zero injections? This number could be anywhere from 15/28 to 21/28 depending upon group overlap. I think it would be beneficial to report the number of patients who were a complete success.

Author Response

1c. My new comment: I do not see any mention regarding the use of a second anti-VEGF or laser therapy in the discussion limitations section.

  • We added this limitation

2c. My new comment: There is no mention of lost to follow up in the results as indicated. Please add this information.

  • we added the data in the results

3c. My new comment: I see no mention of the number of injections per patient in the results, please add.

  • we added the data in the results
  1. My new comment: was there any overlap between the 3 patients with post-operative complications, 3 iatrogenic breaks, and the 7 patients who required corticosteroid therapy after PPV? Of the 28 patients, how many patients had zero complications and needed zero injections? This number could be anywhere from 15/28 to 21/28 depending upon group overlap. I think it would be beneficial to report the number of patients who were a complete success.
  • One of the patients who had a retinal detachment is among those who received two corticosteroid injections. of the 28 patients therefore 16 did not have any intraoperative or postoperative complications and did not need additional treatments during the follow-up. We added it in the results

Reviewer 2 Report

Dear Editor,

Authors have done their best. I don't see any reason to endorse this manuscript.

Kind Regards

Author Response

Thanks for the comments that gave us the opportunity to improve the manuscript.